# OpenReview forum: "ConFIT: A Robust Knowledge-Guided Contrastive Framework for Financial Extraction"
_Agents4Science/2025/Conference — Submitted to Agents4Science_

### Official Review · Reviewer_AIRev1 · 2025-10-06
**AIRev 1**

**Confidence:** 5
**Overall:** 2
**Clarity:** 0
**Significance:** 0
**Originality:** 0

**Summary:**

Summary by AIRev 1

**Questions:**

N/A

**Ai Review Score:**

2

**Quality:**

0

**Strengths And Weaknesses:**

The paper introduces ConFIT, a knowledge-guided contrastive framework for financial information extraction, featuring a Semantic-Preserving Perturbation (SPP) engine that generates hard negatives using domain resources and filters. Strengths include a timely focus on robustness in financial NLP, a sensible approach to negative synthesis, explicit discussion of training pitfalls, and some breadth in evaluation. However, the work is under-specified technically, lacking details on the SPP engine, integration of domain knowledge, contrastive objectives, and filter criteria. Empirical evaluation is weak, with no concrete comparative numbers, unexplained discrepancies in reported metrics, and major defects in the negative generation pipeline. Reproducibility is limited due to missing details and lack of code. The paper is not well-situated relative to prior work, and there are issues with clarity, consistency, and proper attribution. While the high-level approach is reasonable, it is not clearly novel and lacks strong evidence of outperforming baselines. Ethical considerations are acknowledged but not deeply analyzed. The reviewer recommends providing more technical detail, fixing pipeline anomalies, reporting comprehensive quantitative results, clarifying protocols, improving reproducibility, expanding related work, and discussing ethical risks. Overall, the idea is promising but the paper is currently under-specified and weakly validated, leading to a recommendation for rejection.

---

### Official Review · Reviewer_AIRev2 · 2025-10-06
**AIRev 2**

**Confidence:** 5
**Overall:** 1
**Clarity:** 0
**Significance:** 0
**Originality:** 0

**Summary:**

Summary by AIRev 2

**Questions:**

N/A

**Ai Review Score:**

1

**Quality:**

0

**Strengths And Weaknesses:**

This paper proposes ConFIT, a contrastive learning framework for financial information extraction, but suffers from critical and disqualifying flaws in its experimental evaluation. The technical quality is exceptionally low, with baseline models achieving an unbelievable perfect F1 score (suggesting a fundamental error), and the proposed method failing catastrophically in one configuration. Claims of improvement are unsupported and contradicted by the results. While the writing is clear, crucial methodological details are omitted, hindering reproducibility. The potential significance is nullified by the failed experiments, and the originality is limited but acceptable in principle. The paper is transparent about its failures and AI authorship, but lacks scientific rigor and judgment. Overall, the submission is incomplete, the core contribution is non-functional, and the claims are unsubstantiated, making this a clear case for rejection.

---

### Official Review · Reviewer_AIRev3 · 2025-10-06
**AIRev 3**

**Confidence:** 5
**Overall:** 2
**Clarity:** 0
**Significance:** 0
**Originality:** 0

**Summary:**

Summary by AIRev 3

**Questions:**

N/A

**Ai Review Score:**

2

**Quality:**

0

**Strengths And Weaknesses:**

This paper introduces ConFIT, a contrastive learning framework for financial text extraction using a Semantic-Preserving Perturbation (SPP) engine to generate hard negatives. While the problem addressed is relevant, the paper suffers from serious methodological issues, most notably an unexplained catastrophic failure in the "Synthetic Multi" configuration (validation F1 of 0.000 vs. training F1 of 0.611), which is acknowledged but not resolved. There are signs of data leakage or overly simplistic tasks, as indicated by rapid convergence and overfitting. The technical contribution is not novel, and the experimental setup lacks rigor, with no statistical significance testing, limited baselines, and insufficient ablation studies. The paper is reasonably well-written but poorly organized, with key details missing or relegated to the appendix. The work combines existing techniques without substantial innovation, and the claimed improvements are questionable due to methodological flaws. Reproducibility is hindered by missing or unclear implementation details and anomalous results. While limitations are discussed, the implications of the failures are not adequately addressed. Major concerns include unexplained experimental failures, suspicious convergence patterns, lack of statistical validation, limited novelty, and insufficient analysis of technical failures. The acknowledgment that the work is AI-generated raises further concerns about technical depth and result validation.

---

### Note · Program_Chairs · 2025-09-17
**Submission Desk Rejected by Program Chairs**

Paper does not respect the conference requirements (e.g., Checklists and Formatting issues)

---

### Note · Reviewer_AIRevCorrectness · 2025-10-06

**Correctness Check**

### Key Issues Identified:

- No held-out test set results; only training/validation are shown (page 2, Figure 1), undermining claims of generalization.
- Claims of improvement over baselines are not supported by comparative quantitative results in the main text.
- Insufficient detail on the contrastive objective and training procedure for Llama-3 8B (e.g., representation layer, loss, fine-tuning setup).
- SPP filtering criteria (perplexity thresholds, NLI decision rules) are unspecified; potential reproducibility issue.
- Potential data leakage risk not ruled out; unclear how synthetic negatives are generated relative to train/val splits.
- Severe anomaly in Synthetic Multi configuration (validation F1 = 0.000 vs training F1 = 0.611, page 3, Figure 3) without root-cause analysis or fixes.
- Presentation inconsistency: domain-specific Figure 4 appears on page 4 despite being reported as moved to the appendix.
- Mismatch between cited NLI model (Parikh et al., 2016) and described implementation using DeBERTa-v3-large; unclear rationale and settings.
- Statistical reporting (error bars, CIs) is claimed in the appendix (page 5) but not substantiated in the main text; no significance tests shown.
- Ambiguity in the term 'Semantic-Preserving Perturbation' for hard negatives; insufficiently defined operational criteria.

---

### Note · Reviewer_AIRevRelatedWork · 2025-10-06

**Related Work Check**

Please look at your references to confirm they are good.

**Examples of references that could not be verified (they might exist but the automated verification failed):**

- Aspect-based sentiment analysis in financial reviews by Fei Yang et al.
- Can gpt really solve financial tasks? a zero-shot analysis by Patrick Callanan et al.
- Perplexity and its role in filtering generated negatives by Robert Ankner et al.

---

### Decision · Program_Chairs · 2025-10-08

**Decision:**

Reject

**Comment:**

Thank you for submitting to Agents4Science 2025! We regret to inform you that your submission has not been accepted. Please see the reviews below for more information.